# Feral Horses and Bison at Theodore Roosevelt National Park (North Dakota, United States) Exhibit Shifts in Behaviors during Drone Flights

Javier Lenzi [1,*], Christopher J. Felege [1], Robert Newman [1], Blake McCann [2] and Susan N. Ellis-Felege [1]

1 Department of Biology, University of North Dakota, 10 Cornell Street, Stop 9019,
Grand Forks, ND 58201, USA; christopher.felege@und.edu (C.J.F.); robert.newman@und.edu (R.N.);
susan.felege@und.edu (S.N.E.-F.)
2 Theodore Roosevelt National Park, National Park Service, 315 Second Avenue, Medora, ND 58645, USA;
blake_mccann@nps.gov
* Correspondence: javier.lenzi@und.edu

**Abstract:** Drone use has been rapidly increasing in protected areas in North America, and potential impacts on terrestrial megafauna have been largely unstudied. We evaluated behavioral responses to drones on two terrestrial charismatic species, feral horse (*Equus caballus*) and bison (*Bison bison*), at Theodore Roosevelt National Park (North Dakota, United States) in 2018. Using a Trimble UX5 fixed-wing drone, we performed two flights at 120 m above ground level (AGL), one for each species, and recorded video footage of their behaviors prior to, during, and after the flight. Video footage was analyzed in periods of 10 s intervals, and the occurrence of a behavior was modeled in relation to the phase of the flights (prior, during, and after). Both species displayed behavioral responses to the presence of the fixed-wing drone. Horses increased feeding ($p$-value < 0.05), traveling ($p$-value < 0.05), and vigilance ($p$-value < 0.05) behaviors, and decreased resting ($p$-value < 0.05) and grooming ($p$-value < 0.05). Bison increased feeding ($p$-value < 0.05) and traveling ($p$-value < 0.05) and decreased resting ($p$-value < 0.05) and grooming ($p$-value < 0.05). Neither species displayed escape behaviors. Flying at 120 m AGL, the drone might have been perceived as low risk, which could possibly explain the absence of escape behaviors in both species. While we did not test physiological responses, our behavioral observations suggest that drone flights at the altitude we tested did not elicit escape responses, which have been observed in ground surveys or traditional low-level aerial surveys. Our results provide new insights for guidelines about drone use in conservation areas, such as the potential of drones for surveys of feral horses and bison with low levels of disturbance, and we further recommend the development of in situ guidelines in protected areas centered on place-based knowledge, besides existing standardized guidelines.

**Keywords:** behavioral responses; bison; feeding behavior; horse; national parks; protected areas; resting behavior; risk of predation; terrestrial mammals; unmanned aerial vehicles; UAV; vigilance behavior





## 1. Introduction

Over the past decade, drones have contributed to the management of protected areas and species conservation by providing an alternative to traditional aviation and ground surveys. For instance, scientists and practitioners have used this technology to survey and monitor habitat and animal populations [1–4], and assess habitat suitability, structure, and change for a variety of species [5–10]. Additionally, drone technology has provided additional tools for the tracking of radio-tagged animals [11–13], detection of organisms using thermal technology [14–16], and anti-poaching efforts [17,18]. Practitioners use this technology to enforce regulations, conduct environmental management, and improve disaster response protocols [7]. The unique contribution of drones to these pursuits results

from a combination of attributes with advantages and disadvantages as an innovative technology.

One benefit of drones is their flexibility to obtain aerial imagery when and where it is needed, with a reduced financial cost compared to manned aircraft operations. Drones allow surveys to be repeated over time, reducing issues associated with observer fatigue, and they provide extremely high spatial resolution datasets [19–21]. Further, it is broadly assumed that drones are a non-invasive tool with minimal disturbance to wildlife compared to traditional methodologies, and some studies have confirmed this [1,22–24]. However, drones can comprise a novel source of anthropogenic disturbance. A growing body of literature on behavioral responses is revealing that the extent drones impact wildlife depends on the platform employed, flight pattern, and animal-related factors such as life-history stage and behavior [25]. Because behaviors imply the use of time and energy, fitness consequences may emerge in the long term, as well as population limitation by disturbance [26]. Thus, understanding how animal behavior is modified by drones is likely to be relevant to mitigate the negative effects of this novel human activity [27] and to better establish practices that are appropriate for the applications desired, which may range from being minimally invasive to maximally invasive.

Most of the research on behavioral responses of wildlife to drones has been focused on birds [20,22,25,28,29] and marine mammals [1,30–36], with less attention to large terrestrial mammals, especially in North America [37]. For instance, behavioral responses have been studied in black bears (*Ursus americanus*) in North America [38], elephants (*Loxodonta africana*), giraffes (*Giraffa camelopardalis*), wildebeests (*Connochaetes taurinus*), zebras (*Equus quagga*), impalas (*Aepyceros melampus*), lechwes (*Kobus leche*), and tsessebes (*Damaliscus lunatus*) in African savannas [37], kangaroos (*Macropus giganteus*) in Australia [39], captive Przewalski's Horses (*Equus ferus przewalskii*) in China [40], and guanacos (*Lama guanicoe*) in Argentina [41]. All these studies have detected that drone flights trigger behavioral responses, such as an increase in antipredator behaviors, for example, vigilance and escape behaviors. The extent of behavioral changes is likely affected by factors such as flight altitude, noise, speed and the type of drone, time of day, and individual age and sex [25]. Given the limited available literature on behavioral responses of large terrestrial mammals to drones in North America (see Supporting Information S3 in [25]), there is a need to investigate factors that lead to best practices with the use of drone technology that minimize impacts. Additionally, the availability and interest of drones to hobbyists, scientists, and practitioners in protected areas, especially in National Parks and Wildlife Refuges, is increasing globally. Thus, more science to guide the regulation of drone use is needed. In this study, we evaluate if the behaviors of two charismatic species of terrestrial megafauna, feral horses (*Equus caballus*) and bison (*Bison bison*), are modified by the use of a fixed-wing drone in a National Park in the United States. Specifically, we evaluated if behaviors such as feeding, resting, vigilance, traveling, grooming, wallowing, and agonistic changed prior, during, and after drone flights.

## 2. Materials and Methods

This study was carried out in Theodore Roosevelt National Park (North Dakota, United States, Figure 1). The Park comprises 285 km$^2$ of land in three separate units, each having perimeter fencing: The North Unit (97 km$^2$), Elkhorn Ranch Unit (0.9 km$^2$), and South Unit (187 km$^2$), which are linked by the Little Missouri River [42]. The predominant landscape is the North Dakota Badlands, which is an erosive landscape dominated by mixed-grass prairie and sage (*Artemisia* spp.) in upland areas, cottonwood (*Populus deltoides*) woodlands along river corridors, and ash (*Fraxinus pennsylvanica*) draws and juniper (*Juniperus scopulorum*) stands on north-facing slopes. The park also hosts a variety of native Northern Great Plains ungulates, such as bison, elk (*Cervus elaphus*), mule deer (*Odocoileus hemionus*), white-tailed deer (*O. virginianus*), and bighorn sheep (*Ovis canadensis*), among other wildlife species, as well as horses (*Equus caballus*) and longhorn cattle (*Bos taurus*) [42,43]. Horses and cattle are allowed to occur on park lands to demonstrate

a historic scene reminiscent of the open range livestock era experienced by Theodore Roosevelt during his time in the Dakotas. Bison occur in both the North and South Units of the park, and the North Unit herd was analyzed in this study. Horses only occur in the South Unit of the park. Both species interact with humans hiking and touring the park in motor vehicles, and both are exposed to annual or biennial capture events, where excess animals are transferred out of the park to manage population size relevant to available resources on National Park Service lands.

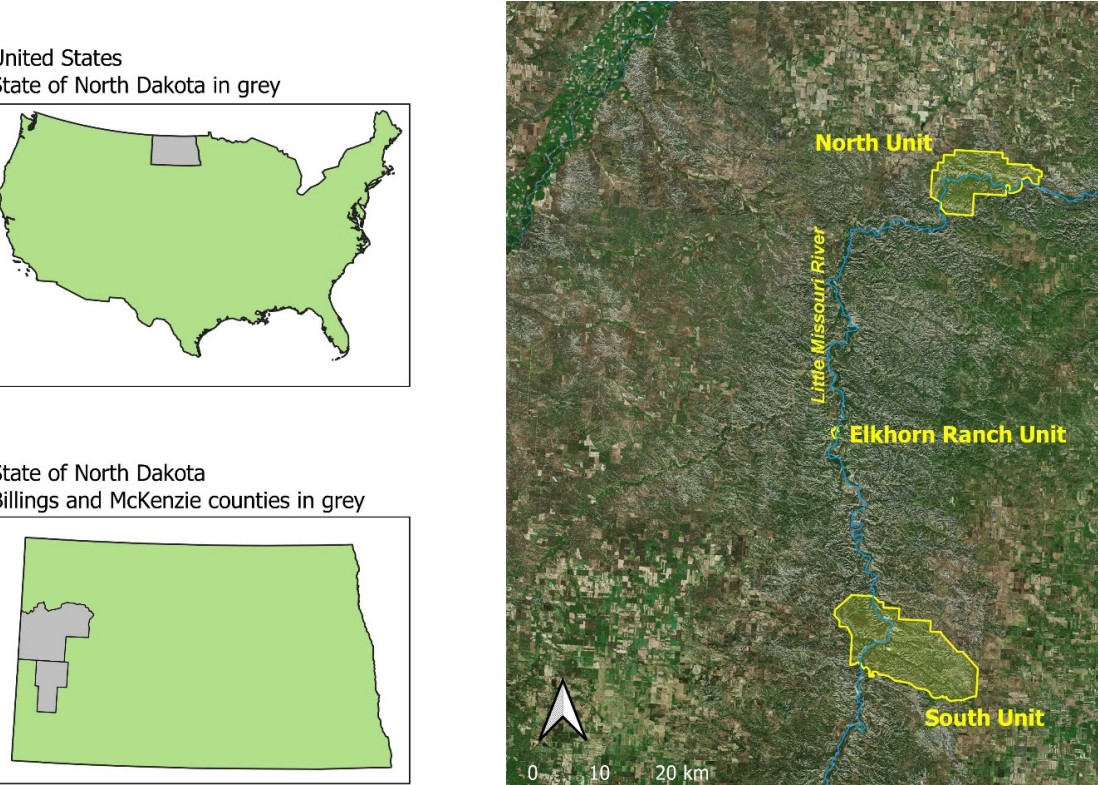

**Figure 1.** Study area at the Theodore Roosevelt National Park in North Dakota, United States.

Our work represents a quasi-experimental design. During the summer of 2018, as part of a larger project surveying vegetation, using a fixed-wing drone at the Theodore Roosevelt National Park, we opportunistically evaluated horse and bison behaviors to drone flights at the North (47°35′32″ N; 103°22′49″ W) and South (46°57′11″ N; 103°28′30″ W) units of the park, respectively. Our evaluations consisted of two flights, one per species, with a Trimble UX5 fixed-wing drone; for a detailed description of this unit, see A. Barnas et al. [22]. The flights were intended to operate directly over the target animals at 120 m (~394 ft) above ground level, below the 122 m (400 ft) maximum ceiling for small UAS under the Federal Aviation Administration of the United States Part 107 operating guidelines. The flight for horses was performed on 27 June 2018 at 15:18 (GMT-6, duration 30 min) and covered a survey area of 0.97 km$^2$. Bison behaviors were evaluated from a flight on 28 June 2018 starting at 14:51 (GMT-7, duration 15 min) and covered a survey area of 0.59 km$^2$. The drone was launched from a catapult at a distance of 621–701 m from horses and 492–551 m from bison. Environmental conditions for the flight over the horses included wind at 13–19 km/h (direction: WNW), air temperature of 26.4 °C, and clear cloud cover. The Bison flight was carried out under wind conditions of 11–13 km/h (direction: SE), air temperature of 30.8 °C, and a cloudy sky. Using a Samsung Galaxy S7 phone camera, from a distance between 150–200 m, we recorded video footage of horse and bison herds prior (horse: 15:50 min, bison: 16:00 min), during (horse: 23:00 min, bison: 15:00 min), and after

(horse: 16:00 min, bison: 15:00 min) each flight (Supplementary Material). While drone flights were conducted in other parts of the park, no other flights had occurred in the area prior to or during the treatments. Unmanned aircraft systems flight operations for this research were approved by the National Park Service (Study# THRO-00099, Permit #THRO-2018-SCI-0010), University of North Dakota animal care (IACUC #1805-3), NPS IACUC (MWR_THRO_McCann_UAS_PrairieDog.Bison.Horse_2016_A3), and University of North Dakota UAS Research Ethics and Privacy (approved 23 May 2018).

The video footage was reviewed in periods of 10 s intervals creating discrete time budget categories following Ransom et al. [44]. During each 10 s unit and following a scan sample approach, behavioral observations were conducted and classified for each species into specific categories prior to, during, and after each flight. Each 10 s interval represents one sample that contains the total number of individuals engaging in a specific behavior relative to the total number of individuals within the field of view. Horse behaviors were classified as follows: feeding, resting, vigilance, traveling, and grooming (Table 1, Supplementary Materials) [44,45]. Bison behaviors were categorized as feeding, resting, traveling, grooming, wallowing, and agonistic (Table 1, Supplementary Materials) [46].

**Table 1.** Classification of horse and bison behaviors video recorded at the Theodore Roosevelt National Park (North Dakota, United States) in 2018. Feral horse behaviors were classified according to [44,45], and bison behaviors were classified following [46].

| Behavior | Description |
|---|---|
| Horse | |
| Feeding | Active foraging with minimum travel |
| Resting | Head hanging, not attentive standing, absence of other behaviors |
| Vigilance | Head above, wither shoulders, and ears forward |
| Traveling | Walking and incidental grazing |
| Grooming | Licking and scratching |
| Bison | |
| Feeding | Active foraging |
| Resting | Standing or laying, head slung |
| Traveling | Active moving without foraging, walking, or running |
| Grooming | Rubbing, scratching with hoof |
| Wallowing | Rolling, dusting, grooming unless urinating and rolling in it |
| Agonistic | Pawing, wallowing with urination, rolling. Usually these behaviors occur in sequence in the presence of two or more males and are more common at the beginning of the breeding season |

To analyze the variation in horse and bison behaviors, we modeled the probability of occurrence of a behavior for an individual within each 10 s interval (trials) as a Bernoulli process. Accordingly, we fitted independent generalized linear models for each behavior, with the occurrence of the behaviors (standardized for the relative number of individuals engaging in that behavior) as the dependent factor; trials were introduced as the independent factor (levels: prior, during, and post flight). A binomial response distribution and logit link function were used for each model. Fitting and analysis were performed using GLIMMIX Procedure of SAS software version 9.4, SAS Institute (Cary, NC, USA).

## 3. Results

The number of 10 s discrete intervals for horses was 325 (prior: 93, during: 138, after: 94) and translated into 4253 individual trials (prior: 1420, during: 1787, and after: 1046). The maximum number of horses monitored in the group during a 10-s period was

25. We monitored an average of 15.3 ± 3.0 individual horses prior to the flight trials, 12.9 ± 5.1 individuals during the flight and 11.1 ± 3.1 individuals after the flight. We evaluated 276 10-s discrete intervals for bison (prior: 95, during: 90, after: 91), which included 6244 recorded trials (prior: 2076, during: 2535, and after: 1633). The maximum number of bison observed during a 10-s discrete period was 36. On average, there were 21.9 ± 1.7 individual bison prior to the flight, 28.2 ± 3.3 individuals during the flight, and 18.1 ± 2.0 individuals after the flight.

Horse feeding behavior significantly increased during the flight (mean occurrence of behaviors relative to the number of individuals engaging in that behavior—hereafter "mean"–mean prior flight: 0.004, mean during flight: 0.161, *p*-value < 0.05) and decreased after (mean after flight: 0.118, *p*-value < 0.05, Figure 2, Table A1). Resting significantly decreased during the flight (mean prior flight: 0.843, mean during flight: 0.115, *p*-value < 0.05) and increased slightly after (mean after flight: 0.271, *p*-value < 0.05, Figure 2, Table A1). Traveling varied across treatment phases, increasing during the flight (mean prior flight: 0.007, mean during flight: 0.135, *p*-value < 0.05) and remained without variation after the flight concluded (mean after flight: 0.137, *p*-value > 0.05, Figure 2, Table A1). Vigilance increased during the flight (mean prior flight: 0.134, mean during flight: 0.586, *p*-value < 0.05) and slightly decreased after, although non-significantly (mean after flight: 0.471, *p*-value > 0.05, Figure 2, Table A1). Grooming decreased during the flight (mean preflight: 0.006, mean during flight: 0.003, *p*-value < 0.05) and continued decreasing after the flight (mean postflight: 0.002, *p*-value < 0.05, Figure 2, Table A1).

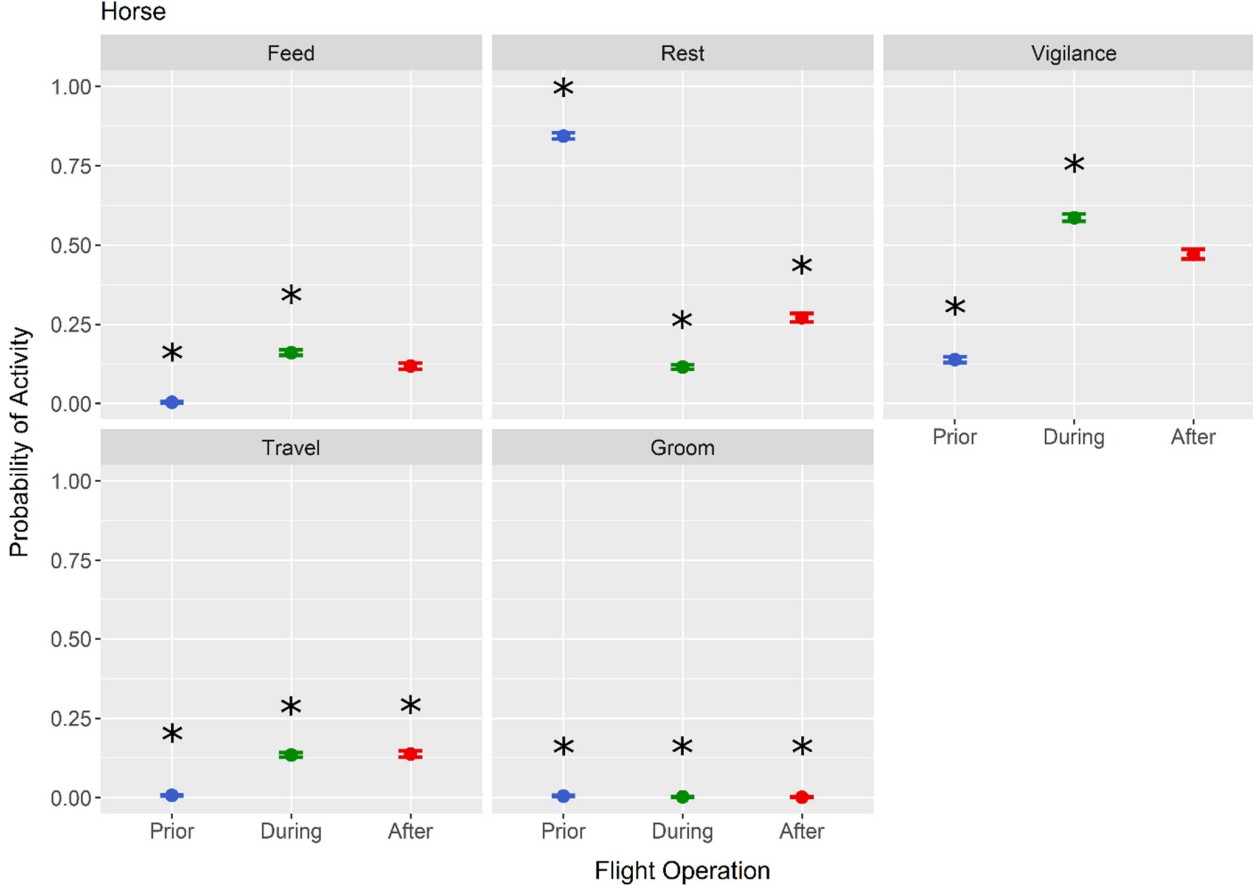

**Figure 2.** Variation in horse behaviors in response to the three stages of the fixed-winged drone treatment: pre, during, and post flight at the Theodore Roosevelt National Park (North Dakota, United States) in 2018. * indicates statistical significance in the estimated parameters (*p*-value < 0.05).

Feeding significantly increased for bison during the flight (mean preflight: 0.375, mean during flight: 0.567, *p*-value < 0.05) and slightly decreased after, although non-significantly (mean postflight: 0.503, *p*-value > 0.05, Figure 3, Table A2). Resting decreased during the flight (mean preflight: 0.564, mean during flight: 0.347, *p*-value < 0.05) and increased after (mean postflight: 0.423, *p*-value < 0.05, Figure 3, Table A2). Traveling significantly increased during the flight (mean preflight: 0.005, mean during flight: 0.008, *p*-value < 0.05) and slightly decreased after the flight concluded (mean postflight: 0.006, *p*-value < 0.05, Figure 3, Table A2). Grooming significantly decreased during the flight (mean preflight: 0.015, mean during flight: 0.006, *p*-value < 0.05) and this behavior was not displayed after the flight (mean postflight: 0.000, *p*-value < 0.05, Figure 3, Table A2). Wallowing and agonistic behaviors were detected in ten and two instances, respectively. Except for one wallowing display, these two behaviors were detected during the preflight phase of the trials. Due to the limited sample sizes, these were not analyzed statistically.

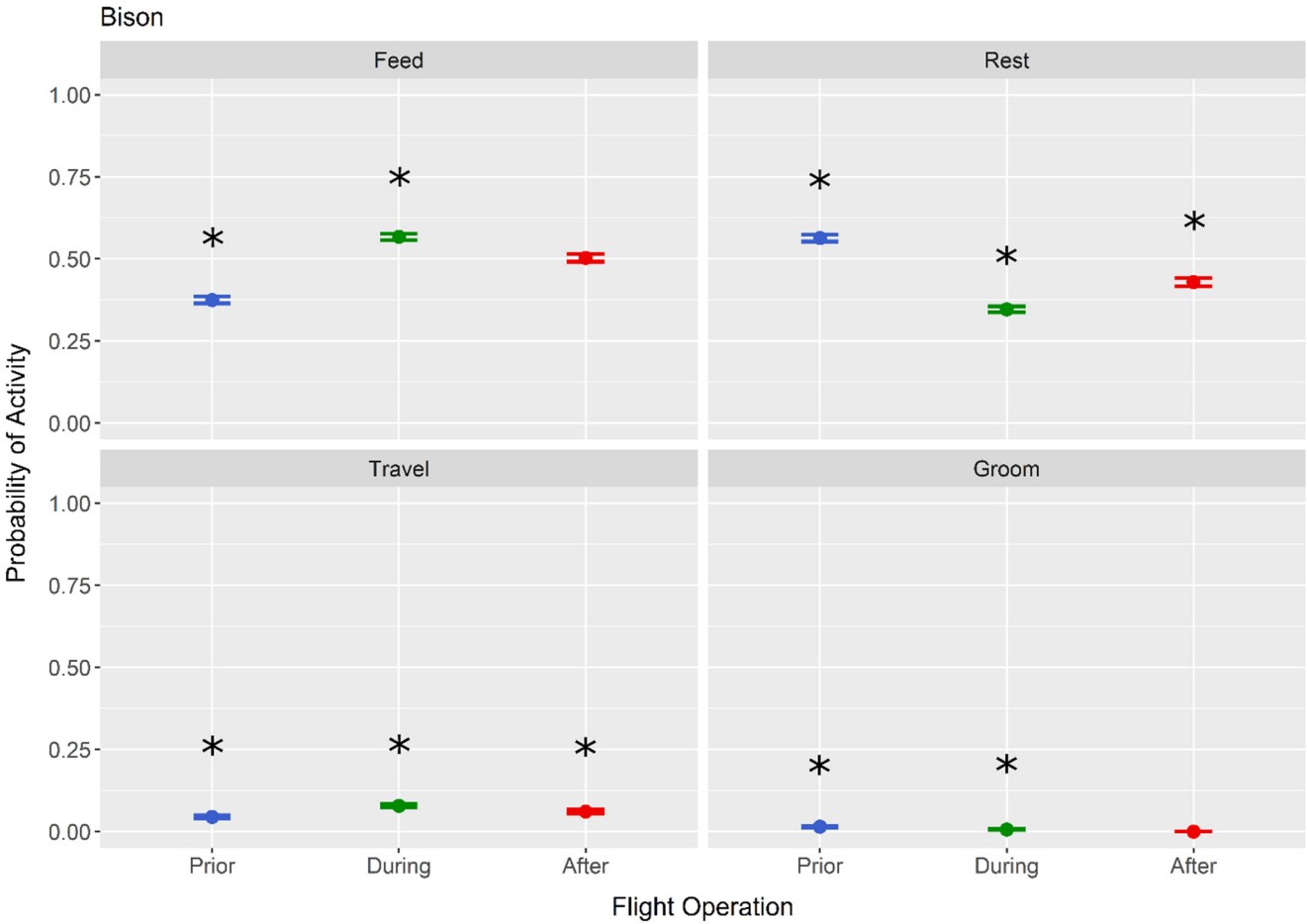

**Figure 3.** Variation in bison behavior in response to the three stages of the fixed-winged drone treatment: pre, during, and post flight at the Theodore Roosevelt National Park in North Dakota, United States, in 2018. * indicates statistical significance in the estimated parameters (*p*-value < 0.05).

## 4. Discussion

To our knowledge, this is the first published study of behavioral responses to drones of bison and feral horses in a conservation area. Both species displayed behavioral responses to the presence of the fixed-wing drone. Horses increased feeding, traveling, and vigilance behaviors, and decreased resting and grooming in response to the drone flight. Bison increased feeding and traveling in response to the drone and decreased resting and grooming behaviors.

The available evidence is not conclusive about the behavioral responses of horses to aerial disturbance, such as drones and manned aircraft, which might be related to the approach and altitude. Using drones, Saitoh and Kobaiaski [47] did not find evidence of behavioral responses of horses to stationary flights, regardless of altitude, in a livestock management context. Moreover, McDonnell and Torcivia [48] conducted a proof of concept study for gathering and handling wild semi-feral ponies using a drone to lead but not chase individuals. These authors finally exploited the curiosity of these ponies because the herd followed the drone to specific destinations. Possibly, ponies could be attracted to the drone given the low altitude of the approaches (2–6 m above ground level) and the short horizontal distance (10 m) ahead of the herd, and the flight approach that was gradual and intermittent with hovering and sweeping retreats. Further, evidence from assessments of manned aircraft as sources of disturbance provides inconsistent conclusions. For instance, using helicopters Hansen and Mosley [45] did not find differences in foraging and social behaviors of feral horses when compared between treatments (individuals gathered by helicopter) versus control (individuals not gathered by helicopter) groups. However, Linklater and Cameron [49] found that the escape behavior of individual feral horses was triggered using helicopters from horizontal distances between 1 and 2.75 km. Thus, researchers and practitioners could be able to finetune flight plans and use specific aircraft to meet certain needs, such as leading, dispersing, or triggering escape behaviors of horses.

Evidence about behavioral responses of bison to human disturbance was developed in the context of individual reactions to terrestrial human approaches, with less research on aerial disturbance. For example, Fortin and Andruskiw [50] studied behavioral responses of bison to human presence and found that vigilance and fleeing were the most common reactions. Borkowski et al. [51] analyzed bison responses to winter recreation in Yellowstone National Park. These authors found that individual bison tended to increase alert, travel, and escape behaviors when groups were closer to roads and when there were fewer animals in the group. Further, bison appeared to habituate when the number of daily vehicles entering the park increased [52]. Finally, Anderson [53] reviewed the effect of manned aircraft disturbance on wildlife and found that only 2 out of 59 groups of bison responded (although the author did not describe specific behaviors) to aircraft overflights and suggested that herds might be habituated to this disturbance. Our observations supported the idea of fewer escape behaviors for overflight than with human presence on the ground and a similar response of limited reactions to aircraft.

The drone used in this study might have been perceived by horses and bison as low risk, possibly given the high altitude and small drone size. Our results showed that individuals did not display escape behavior, and even increased feeding activities in response to drone flights at 120 m above ground level. Evidence from other studies using drones as a source of disturbance shows that escape behavior is typically the normal way that terrestrial mammals respond to drones, but altitude is likely a key factor. For example, Przewalski's Horses reacted to drones at different altitudes eliciting escape behavior when flights are below 20 m above ground level [40]. Another study in Argentina showed that guanacos elicited escape behavior in response to drones and the escape distance decreases with the size of the group, with the highest probability of reaction at 60 m AGL [41]. In the African savanna, elephants, giraffes, wildebeests, zebras, impalas, lechwes, and tsessebes showed escape behavior to drones at altitudes between 20 and 60 m AGL [37]. Brunton et al. [39] showed that kangaroos responded to drone flights by increasing vigilance but did not display escape behavior at altitudes higher than 30 m; below this altitude, escape behavior could be triggered. In our study, the drone flight was performed higher than those studies that recorded escape behaviors in response to drones, supporting the idea that at this altitude, drones might not be perceived as a threat by large mammals, such as horses and bison. Moreover, the fact that feeding activities increased in response to the presence of the drone supports the idea that perhaps the drone is noticed but not perceived as a threat for both species to leave the area. In addition, roundups in the park have been conducted

using helicopters for horses (last horse roundup in 2013) and bison (last bison roundup in 2016), which could be perceived as high risk. Comparatively, our drone could have been perceived as a lower-risk threat than helicopters. Nevertheless, after a stress event, a bison was observed in our study area, feeding nervously after running up to the observer and pawing. Based on this observation, it could be possible that bison and horses have elicited displacement behaviors [54,55] towards feeding when running and pawing were unsuccessful. Thus, drones could be perceived as low risk, but also, we might consider displacement towards feeding as a mechanism to deal with stress, reducing occurrences of escape behaviors, as well.

Individuals not displaying escape behaviors instead of fleeing in response to drone flights is not necessarily a signal of the absence of impact. Conversely, alternative responses such as physiological stress have been suggested to occur in response to drone flights, indicating the need to further account for this type of response. For instance, Ditmer [38] analyzed the effect of drones on the heart rates of American black bears, showing that individuals increased heart rates in response to drone flights at close proximity. Using manned aircraft, MacArthur et al. [56] also found that heart rates of mountain sheep (*Ovis canadensis canadensis*) increased in response to helicopter overflights at distances less than 400 m. In the case of some birds, they will hold still as drones fly over, and while they are not escaping, they could have physiological responses in preparation for escape occurring [57]. Therefore, an understanding of the physiological responses of wildlife to drones needs to be developed by future studies [57] because drones are a novel source of human disturbance, with the potential to induce more frequent stress in the near future, as their use is increasing by scientists and hobbyists [38].

Although this study is preliminary and was performed with a limited dataset, our results provide new insights for guidelines about drone use in conservation areas. This study suggests that drones could be perceived as low risk, indicating they may serve as an appropriate tool for surveys of these species with low levels of disturbance, unlike other stressful sources. Recent works, such as Hodgson and Koh [58] and Mulero-Pázmány et al. [25], have suggested a number of standards to minimize the impacts of drones on wildlife by selecting small and silent drone units and experienced pilots, as well as considering the best practices during operations (e.g., perform short missions, fly as high as possible, do not maneuver over wildlife, and monitor target individuals before, during, and after the flight). However, as our study also suggests, not all species and individuals respond the same way to drone disturbance and are not under the same risk of predation, which could also change over time and space and be affected by the type of drone or operational conditions. Thus, the efforts in developing broad standards to regulate the use of drones in natural areas and elsewhere should persist, but we also recommend the development of in situ guidelines in protected areas, centered on place-based knowledge about the behavioral responses of local animal communities.

## 5. Future Work

Evidence about horse and bison behavioral responses to disturbance appears to be fragmented and unstandardized, making comparisons difficult. For example, although studies are scarce, they are also diverse in their use of comparative axes (e.g., groups and individuals), environmental settings (e.g., in captivity and in the wild), they record only a small set of behaviors (mostly escape behavior), and there is almost no information on aerial disturbances for bison. Considering the limited amount of literature about behavioral responses in these species, we recommend that future studies consider standardized experimental designs that allow for comparisons at the group and individual levels. For instance, behavioral responses of groups might consider designs allowing mean comparisons across flight phases (like this study), and responses of individuals might consider repeated measures designs. Additionally, it is recommended to expand the datasets gathered by this study incorporating other factors, such as time of day, seasonal and annual cycles, food availability, among others, and to fit models considering autoregressive error structures

to account for temporal interdependence of behaviors, or other approaches. Further, this type of analysis could be expanded to other large mammals in protected areas and beyond. Additionally, to have a better understanding of how time budgets are modified by drones, it is advisable to incorporate behaviors from ethograms instead of analyzing a limited number of behaviors, such as alert and escape. Because the absence of escape and alert behaviors do not necessarily indicate the absence of impact, it could be beneficial to incorporate animal tagging technology to account for the physiological responses of horses and bison to drones. Finally, to analyze the possible effect of the species identity on the mammal behavioral responses to drones, multispecies assessments could be informative.

**Supplementary Materials:** Video footage illustrating horse and bison behaviors can be obtained at https://vimeo.com/showcase/9469204 (accessed on 23 May 2022).

**Author Contributions:** Conceptualization, S.N.E.-F. and C.J.F.; Methodology, S.N.E.-F. and C.J.F.; data curation, S.N.E.-F. and C.J.F.; formal analysis, S.N.E.-F., C.J.F. and J.L.; visualization, J.L. and S.N.E.-F.; writing—original draft preparation, J.L., S.N.E.-F. and C.J.F.; writing—review and editing, J.L., S.N.E.-F., C.J.F., R.N. and B.M.; supervision, S.N.E.-F. All authors have read and agreed to the published version of the manuscript.

**Funding:** This research was supported by the University of North Dakota Postdoctoral Seed Grant Program, UND Program #02703.

**Institutional Review Board Statement:** Not applicable.

**Informed Consent Statement:** Not applicable.

**Data Availability Statement:** The behavioral dataset can be obtained upon request.

**Acknowledgments:** We wish to thank the University of North Dakota Biology Department and College of Arts and Sciences for their support, staff at Theodore Roosevelt National Park and Aviation Safety. J. Folluo, M. Thompson, and J. Irving provided additional assistance during flight operations. We thank B. Darby for statistical recommendations on the analysis.

**Conflicts of Interest:** The authors declare no conflict of interest.

## Appendix A

**Table A1.** Parameter estimation of generalized linear models for horse behavior responses to drone flight trials (pre, during, and post flights) at the Theodore Roosevelt National Park (North Dakota, United States) in 2018.

|  | Estimate | Standard Error | *p*-Value |
|---|---|---|---|
| Feeding |  |  |  |
| Pre | −5.46 | 0.41 | <0.001 |
| During | −1.65 | 0.06 | <0.001 |
| Post | −2.00 | 0.09 | <0.001 |
| Resting |  |  |  |
| Pre | 1.68 | 0.07 | <0.001 |
| During | −2.03 | 0.07 | <0.001 |
| Post | −0.99 | 0.06 | <0.001 |
| Vigilance |  |  |  |
| Pre | −1.82 | 0.07 | <0.001 |
| During | 0.34 | 0.04 | <0.001 |
| Post | −0.11 | 0.06 | 0.065 |

**Table A1.** *Cont.*

|  | Estimate | Standard Error | *p*-Value |
|---|---|---|---|
| Traveling |  |  |  |
| Pre | −4.85 | 0.30 | <0.001 |
| During | −1.85 | 0.06 | <0.001 |
| Post | −1.83 | 0.08 | <0.001 |
| Grooming |  |  |  |
| Pre | −5.17 | 0.35 | <0.001 |
| During | −5.69 | 0.40 | <0.001 |
| Post | −6.25 | 0.70 | <0.001 |

**Table A2.** Parameter estimation of generalized linear models for bison behavioral responses to drone flight trials (pre, during, and post flights) at the Theodore Roosevelt National Park (North Dakota, United States) in 2018.

| Behavior | Estimate | Standard Error | *p*-Value |
|---|---|---|---|
| Feeding |  |  |  |
| Pre | −0.51 | 0.05 | <0.001 |
| During | 0.27 | 0.04 | <0.001 |
| Post | 0.01 | 0.05 | 0.786 |
| Resting |  |  |  |
| Pre | 0.26 | 0.04 | <0.001 |
| During | −0.63 | 0.04 | <0.001 |
| Post | −0.28 | 0.05 | <0.001 |
| Traveling |  |  |  |
| Pre | −3.06 | 0.11 | <0.001 |
| During | −2.46 | 0.07 | <0.001 |
| Post | −2.73 | 0.10 | <0.001 |
| Grooming |  |  |  |
| Pre | −4.19 | 0.20 | <0.001 |
| During | −5.06 | 0.22 | <0.001 |
| Post | −19.66 | 458.93 | 0.966 |

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
