# Peer review of "Feral Horses and Bison at Theodore Roosevelt National Park (North Dakota, United States) Exhibit Shifts in Behaviors during Drone Flights"

_drones, doi:10.3390/drones6060136_

Round 1

Reviewer 1 Report

Dear authors,

I have found your work interesting and worth publication. I only recommend to describe at the end of the introduction section, at least in general terms, what kind of behaviours are you going to study.

Author Response

Reviewer 1:

Dear authors,

I have found your work interesting and worth publication. I only recommend to describe at the end of the introduction section, at least in general terms, what kind of behaviours are you going to study.

Response: Thank you for reviewing our manuscript in such a timely fashion. We have incorporated the type of behaviors that were analyzed at the end of the introduction.

Reviewer 2 Report

The title accurately reflects the major findings of the work.

The abstract adequately summarize methodology, results, and significance of the study.

However, Authors should indicate the statistical analysis and the statistical results including the P values.

Keywords represent the article adequately.

The introduction section is well written and it falls within the topic of the study. Appropriate bibliographic references are cited and appropriate information on the state of the art on this topic.

I suggest to construct the sentences in third person throughout the manuscript.

The section of Materials and Methods is clear for the reader and it meticulously describes the methods applied in the study.

Results section as well as Discussion section is clear and well written. The findings obtained in the study were well discussed and justified with appropriate references.

I would like to congratulate the authors for the figures which, to my opinion, are very nice and well represent the results of the study.

The paragraph (Lines 278-296) should be simplified in order to became a conclusion section in which the Authors summarize the results and the significance of the study.

Authors should check and standardize the references in the list according to journal guidelines.

Author Response

The title accurately reflects the major findings of the work.

The abstract adequately summarize methodology, results, and significance of the study.

However, Authors should indicate the statistical analysis and the statistical results including the P values.

Response: Thank you very much for having reviewed our manuscript. We have incorporated the P-values in the abstract as suggested.

Keywords represent the article adequately.

The introduction section is well written and it falls within the topic of the study. Appropriate bibliographic references are cited and appropriate information on the state of the art on this topic.

I suggest to construct the sentences in third person throughout the manuscript.

Response: Thank you for this suggestion. While we have changed the way we describe things in some places, we have chosen to keep it in first person as that lends itself to the recommended active voice style.

The section of Materials and Methods is clear for the reader and it meticulously describes the methods applied in the study.

Results section as well as Discussion section is clear and well written. The findings obtained in the study were well discussed and justified with appropriate references.

I would like to congratulate the authors for the figures which, to my opinion, are very nice and well represent the results of the study.

The paragraph (Lines 278-296) should be simplified in order to became a conclusion section in which the Authors summarize the results and the significance of the study.

Response: Thanks for this suggestion. We have simplified the last paragraph eliminating examples and citations. 

Authors should check and standardize the references in the list according to journal guidelines.

Response: Thanks for this suggestion. Although we have used a reference manager, we have reviewed the reference list and found a few typos that were solved. We hope reference list is clean in this new version.

Reviewer 3 Report

Review of drones-1719362

This is an interesting paper that tests the impact of UAV flight altitude to the behavioral responses of two large terrestrial mammals, feral horses and bison.

INTRODUCTION

The introduction clearly states the problem and provides a thorough assessment of the existing literature.

Lines 74-78. At the end of the introduction, the authors need to clearly present their predictions, based on the available literature on other large terrestrial mammals. Please amend. 

MATERIAL AND METHODS

Line 80 and thereafter. Please provide coordinates of the park, unit, or the exact study area.

Line 88. Please add the Latin binomial of the bison (Bison bison) in italics.

Lines 102-121. The experimental design seems quite solid but there are a few things that seem awkward. Why have not the authors used the same area for testing the drone flights, e.g., the south unit area where both horses and bison occur syntopically? In this way, ground environmental conditions would have been rather similar, eliminating biases. Why was not the experiment repeated the next day in each study group to collect more behavioral data and strengthen their sample and test for behavioral idiosyncrasies or abnormalities? The VERY short periods of sampling prior, during and after drone flight simply provide a glimpse of behavioral changes and do not Ten seconds appear as a short time unit to collect data on large terrestrial mammals as successive behaviors would be interdependent. The authors tested for that via their models? It is not clear if they authors collected data using scan sample (all the animals visible in the camera) or focal sample? Please specify and explain the choice. Judging by the results they used scan sampling methods but this needs to be clarified in the methods section.

Table 1. Bison did not exhibit any vigilance behaviors?

RESULTS

Very well and clearly presented.

DISCUSSION

Overall, the discussion is well articulated and specifies and analyzes the limitations of the study.  It should be important to note that the present study is preliminary and attempts to draw some guidelines for future work on the impact of drones to wildlife. Although the sample was very limited and occurred for a very specific experimental goal it provided some hints that can help understand how and why drones affect wildlife observation and what can be done to minimize their effects while keeping the benefits they offer for conservation and behavioral studies. In this way it should try to investigate drone parameters (e.g., shape, noise, flight altitude, operational methods, maneuvering abilities, etc.) to wildlife variables (e.g., habitat conditions, season, species, age-sex groups, etc.). The authors have adequately tackled these issues rendering their short experiment into a valuable tool for further work.

Author Response

Review of drones-1719362

This is an interesting paper that tests the impact of UAV flight altitude to the behavioral responses of two large terrestrial mammals, feral horses and bison.

Response: Thank you for taking the time to review our manuscript.

INTRODUCTION

The introduction clearly states the problem and provides a thorough assessment of the existing literature.

Lines 74-78. At the end of the introduction, the authors need to clearly present their predictions, based on the available literature on other large terrestrial mammals. Please amend.

Response: Thank you very much for this suggestion. In this research we opted for not following the Hypothetico-deductive Method because to use this method we should have had conducted a rigorously controlled experiment, where the only possible source of causality for variation in the variable that is being measured, besides the intrinsic variation (sampling error), is the phenomenon pointed out by the scientific hypothesis (Quinn & Dunham 1983, Camus & Lima 1995). Although our design is a quasi-experiment we could not control for any environmental, psychological, or other factor, so it would be unfair to test a scientific hypothesis in these conditions. This point is also related to questions below in relation to having more measurements of the herds in different contexts. We had limited chances to work with these herds and performed the best possible design within an opportunistic situation for gathering data.

Quinn JF & AE Dunham (1983) On hypothesis testing in ecology and evolution. American Naturalist 122: 602-617.

Camus PA & M Lima (1995) El uso de la experimentación en ecología: Fuentes de error y limitaciones. Revista Chilena de Historia Natural. 68: 19-42.

MATERIAL AND METHODS

Line 80 and thereafter. Please provide coordinates of the park, unit, or the exact study area.

Response: Thanks for this observation. We have provided the coordinates of the North and South units in the text.

Line 88. Please add the Latin binomial of the bison (Bison bison) in italics.

Response: Thanks for this observation. Bison was first named in line 76 and the scientific name is in italics.  As a result, we did not include it again here.

Lines 102-121. The experimental design seems quite solid but there are a few things that seem awkward. Why have not the authors used the same area for testing the drone flights, e.g., the south unit area where both horses and bison occur syntopically? In this way, ground environmental conditions would have been rather similar, eliminating biases.

Response: Thank you very much for this observation. While both species do occur in the south unit, our study used opportunistic observations where the animals were occurring with other study objectives focused on habitat assessments and where we had pre-approved our flights to not disrupt visitors and other park activities.

Why was not the experiment repeated the next day in each study group to collect more behavioral data and strengthen their sample and test for behavioral idiosyncrasies or abnormalities?

Response: This is a good question. This study was opportunistic and focused on the initial behavioral responses of the animals to the novel stimuli rather than repeated observations.  If multiple herds were available and accessible for sampling, additional samples would have definitely improved our study. Unfortunately, that was not possible. We hope this could be reflected along the manuscript, specifically along lines 143-148 and in line 352 of the revised version of the manuscript.

The VERY short periods of sampling prior, during and after drone flight simply provide a glimpse of behavioral changes and do not Ten seconds appear as a short time unit to collect data on large terrestrial mammals as successive behaviors would be interdependent. The authors tested for that via their models?

Response: We agree with the reviewer that our analysis provides a glimpse of the behavioral changes. No, we have not tested for this interdependence in the models. While the 10-second intervals are short, they were used as way to determine behaviors across a number of individuals and there were more than 90 of the 10-second intervals for each period since each period was at least 15 minutes in duration. However, although our dataset is limited, the number of behaviors recorded within the 10-second intervals was large enough to perform simple comparisons between flight phases. A more robust sampling design certainly would have allowed further evaluation of interdependence of behaviors within each of the flight phases, and modelling considering autoregressive error structures or other approaches. We tried to make this explicit in lines 391-398 of this new version: “Additionally, it is recommended to expand the datasets gathered by this study incorporating other factors such as time of day, seasonal and annual cycles, food availability, among others, and to fit models considering autoregressive error structures to account for temporal interdependence of behaviors”.

It is not clear if they authors collected data using scan sample (all the animals visible in the camera) or focal sample? Please specify and explain the choice. Judging by the results they used scan sampling methods but this needs to be clarified in the methods section.

Response: Thank you for this observation. We have made this clearer that we were using a scan sampling approach.  See revised manuscript between lines 176-179: “During each 10 second unit and following a scan sample approach, behavioral observations were conducted and classified for each species into specific categories prior to, during, and after each flight.”

Table 1. Bison did not exhibit any vigilance behaviors?

Response: Thank you for this question. Actually, bison not displaying increased vigilance was a question that raised among authors. Our thoughts are that bison body structure is not really conducive to looking overhead (and there are probably not many overhead predators for animals of this size) so this prevented us to capture this behavior. In addition, we probably were not close enough to pick up what were probably minor movements, even eye movements, that might have been classified as vigilance.

RESULTS

Very well and clearly presented.

DISCUSSION

Overall, the discussion is well articulated and specifies and analyzes the limitations of the study.  It should be important to note that the present study is preliminary and attempts to draw some guidelines for future work on the impact of drones to wildlife. Although the sample was very limited and occurred for a very specific experimental goal it provided some hints that can help understand how and why drones affect wildlife observation and what can be done to minimize their effects while keeping the benefits they offer for conservation and behavioral studies. In this way it should try to investigate drone parameters (e.g., shape, noise, flight altitude, operational methods, maneuvering abilities, etc.) to wildlife variables (e.g., habitat conditions, season, species, age-sex groups, etc.). The authors have adequately tackled these issues rendering their short experiment into a valuable tool for further work.

Response: Thank you for the positive comments.  Based on your response, it appears we were able to successfully articulate the goals of our limited study to help expand future work.